# Impact of Geriatric Nutritional Risk Index and Modified Creatinine Index Combination on Mortality in Hemodialysis Patients

**DOI:** 10.3390/nu14040801

**Published:** 2022-02-14

**Authors:** Hayato Fujioka, Tsutomu Koike, Teruhiko Imamura, Fumihiro Tomoda, Kota Kakeshita, Hidenori Yamazaki, Koichiro Kinugawa

**Affiliations:** 1The Second Department of Internal Medicine, Toyama University, Toyama 930-0194, Japan; hfujioka@med.u-toyama.ac.jp (H.F.); tkoike@med.u-toyama.ac.jp (T.K.); kakeshit@med.u-toyama.ac.jp (K.K.); yamazaki@med.u-toyama.ac.jp (H.Y.); kinugawa@med.u-toyama.ac.jp (K.K.); 2Faculty of Health Science, Fukui Health Science University, Fukui 910-3190, Japan; tomoda@fukui-hsu.ac.jp

**Keywords:** malnutrition, inflammation, end-stage renal disease

## Abstract

The prognostic impact of the combination of a geriatric nutritional risk index (GRNI) and modified creatinine index, both of which assess nutritious status in hemodialysis patients, has not yet been well investigated thus far. Patients receiving maintenance hemodialysis in our institutes between February 2011 and January 2017 were retrospectively included. The baseline GRNI and modified Creatinine index were calculated and the impact of their combination on 5-year all-cause mortality following the index hemodialysis was investigated. A total of 183 patients (68.3 ± 12.4 years, 98 men, hemodialysis duration 97 ± 89 months) were followed from the index hemodialysis for 5.5 years. Mean GNRI was 91.2 and mean modified Creatinine index was 22.2 in men and 19.6 in women. The 5-year survival was significantly stratified by the median values of GNRI and modified Creatinine index (*p* < 0.05 for both). Patients with low GNRI and a low modified Creatinine index had lower 5-year survival than those with the other three combination patterns (*p* < 0.05). A combination of GNRI and modified Creatinine index may be a promising tool to risk stratify mortality in dialysis patients.

## 1. Introduction

Hemodialysis patients often have malnutrition and chronic inflammation, both of which synergistically progress end-organ dysfunction and atherosclerotic diseases [1,2]. This has recently been called malnutrition-inflammation-atherosclerosis syndrome. Of note, the unique malnutrition status in patients with hemodialysis, focusing on reduced dietary intake, chronic inflammation, resistance to anabolic hormones, loss of amino acids via dialysate, and muscle protein breakdown by hemodialysis, is called protein-energy wasting [3]. Protein-energy wasting is associated with mortality and morbidity as well as an impaired quality of life in hemodialysis patients [4,5,6].

Several indexes to assess protein-energy wasting, including geriatric nutritional risk index (GNRI) and modified creatinine (Cr) index, have been introduced [6,7,8,9,10,11]. GNRI is a simple tool for assessing nutritional status in various pathological conditions based on body mass index and serum albumin levels, focusing on visceral protein assessment [7]. Modified Cr index is a tool that reflects creatinine production, that is, skeletal muscle mass, focusing on the assessment of somatic protein [8]. Both have been independently shown to be associated with an increased risk for mortality and cardiovascular death in patients with hemodialysis. However, the prognostic impact of a combination of these two indexes remains uncertain. Given that the focuses of these indexes are independent (visceral protein status versus somatic protein status), we hypothesized that a combination of both indexes would have a further valuable prognostic impact on hemodialysis patients.

## 2. Materials and Methods

### 2.1. Patient Selection

Consecutive patients who continued standard maintenance hemodialysis in our institute and associated dialysis centers between February 2011 and January 2017 were retrospectively included. None of them were on special diets. Patients aged under 20 years old or with missing data were excluded. The institutional ethical review board approved the study protocol. The informed consent was wavered given the retrospective nature of this study and the opt-out of this study protocol.

### 2.2. Baseline Data Collection

Baseline demographics, comorbidity, vital signs, medication data, and laboratory results were obtained from medical records. Comorbidity was also assessed using the Charlson risk index [12]. Blood urea nitrogen was measured before and after the index hemodialysis to determine Kt/V for urea, which was calculated using the Daugirdas method.

### 2.3. Index Calculation

GNRI was calculated using serum albumin level and body size as follows: [14.89 × serum albumin (g/dL)] + {41.7 × [current body weight (kg)/standard body weight (kg)]} [7]. Standard body weight was calculated as height (m)^2^ × 22. If the current weight was greater than the standard weight, we set the ratio of current to standard weight = 1.

Modified Cr index was calculated as follows: 16.21 + 1.12 × (1 if male; 0 if female) − 0.06 × [age (years)] − 0.08 × (Kt/V for urea) + 0.009 × [serum Cr (μmol/L)] [8].

### 2.4. Follow-Up

All patients were followed from the index hemodialysis until the end of study period (January 2017). The primary outcome was all-cause mortality. The secondary outcome included death due to cardiovascular diseases (cardiac arrest, heart failure, coronary heart disease, aortic dissection and a cerebrovascular accident, including cerebral bleeding or infarction), infectious disease (respiratory, urinary tract, intestinal, cardiac, neurologic, soft tissue, septicemia, vascular access-related, and others), and cancer (solid and hematologic).

### 2.5. Statistics

Data are expressed as mean ± standard deviation for continuous variables and the number and percentage for categorical variables. The patient cohort was stratified by the median levels of GNRI and modified Cr index, respectively. The freedom from the primary/secondary endpoints was stratified by the two indexes and compared using a log-rank test. The time-to-event analysis was performed using Cox proportional hazard ratio regression analysis, which was adjusted for clinically significant variables including age, sex, dialysis vintage, diabetes mellitus, history of cardiovascular events, Charlson risk index, hemoglobin, phosphate, total cholesterol, C-reactive protein, and intact parathormone. Two-tailed *p* < 0.05 was considered statistically significant. Analyses were performed using R software version 3.5.2 (R Foundation for Statistical Computing, Vienna, Austria).

## 3. Results

### 3.1. Baseline Characteristics

A total of 193 patients were considered for inclusion in this study. Of them, 10 were excluded due to missing data. Finally, 183 patients (68.3 ± 12.4 years, 98 men, hemodialysis duration 97 ± 89 months) were included (Table 1). Mean GNRI was 91.2 ± 10.9. Mean modified Cr index was 22.2 ± 2.5 in males and 19.6 ± 2.2 in females, respectively. The causes of end-stage renal disease were as followed; 51 chronic glomerulonephritis, 76 diabetic nephropathy, 18 nephrosclerosis, 16 polycystic kidney disease, and 22 others.

### 3.2. Clinical Outcomes

During a mean follow-up period of 5.5 years, there were 70 deaths (28 cardiovascular diseases, 21 infectious diseases, 11 cancer, and 10 others).

### 3.3. Stratification of Clinical Outcomes by GNRI

Patients were divided into two groups based on the median GNRI, namely the higher group (GNRI ≥ 91.6, *n* = 93) and the lower group (GNRI < 91.6, *n* = 90). The five-year survival was significantly lower in the lower GNRI group (45% versus 76%, *p* < 0.001; Figure 1A). The lower GNRI group was associated with the incidence of primary endpoint in all three multivariate models including potential confounders (*p* < 0.05 for all; Table 2).

Freedom from death due to cardiovascular and infectious diseases was also significantly stratified by the GNRI levels, respectively (*p* < 0.05 for both), whereas those of malignant disease were not statistically different (*p* = 0.70).

### 3.4. Stratification of Clinical Outcomes by Modified Cr Index

Patients were also divided into two groups based on the median modified Cr index of each gender, i.e., the higher group (modified Cr index ≥22.3 in male or ≥19.9 in female, *n* = 92) and the lower group (modified Cr index <22.3 in men or <19.9 in female, *n* = 91). The lower Cr index group had significantly lower 5-year survival (45% versus 76%, *p* < 0.001; Figure 1B). The modified Cr index was associated with the incidence of primary endpoint only in model 1 (*p* < 0.05; Table 2).

The modified Cr index stratified the incidence of cardiovascular death and infection, respectively (*p* < 0.05 for both), whereas mortality due to malignancy was not stratified by modified Cr index (*p* = 0.33).

### 3.5. Stratification of Clinical Outcomes by Both Indexes

Fifty-nine patients had low GNRI and low modified Cr index, and 54 had high GNRI and high modified Cr index. Other 70 patients had either low GNRI or low modified Cr index (Table 3).

A combination of low GNRI and low modified Cr index was associated with a lower 5-year survival compared with the other three combination patterns (32%, *p* < 0.001 for all; Figure 1C). Among the other three combination patterns, 5-year survivals were statistically not different (*p* > 0.05 for all). Similar trends were observed in cardiovascular death and infection death (Figure 2A,B) but not in cancer death (*p* = 0.22; Figure 2C). Cox proportional hazard ratio regression analyses showed trends similar to the findings of log-rank test (Table 4).

## 4. Discussion

In this study, we investigated, for the first time, the impact of a combination of two major malnutrition indexes, GNRI and modified Cr index on mortality in patients receiving maintenance hemodialysis. The 5-year survival was significantly stratified by the median values of GNRI and modified Cr index, respectively. Fifty-nine patients had low GNRI and a low modified Cr index, whereas 70 had either of them. Patients with low GNRI and low modified Cr index had lower 5-year survival than those with the other three combination patterns.

### 4.1. GNRI and Modified Cr Index

Yamada and colleagues recently demonstrated, using the J-DOPPS registry data, that the ability to predict survival in hemodialysis patients was statistically comparable between GNRI and modified Cr index [13]. The association between the two indexes was discussed in this study. A worse GNRI was associated with low serum Cr level, which is a dominant component of the modified Cr index. On the contrary, a worse modified Cr index was associated with low serum albumin level, which is a dominant determinant of GNRI. Both indexes were equally associated with body mass index and normalized protein catabolic rate, both of which are reliable markers of nutritious status in patients receiving hemodialysis. Therefore, they concluded that both indexes assess, at least partially, a similar nutritious status. However, they did not investigate the prognostic impact of the combination of these two indexes.

We hypothesized that a combination of GNRI and modified Cr index can be used to predict the survival of hemodialysis patients. In our study, we observed that GNRI and modified Cr index were not completely parallel. Approximately 40% of the patients had either a low GNRI or low modified Cr index. GNRI and modified Cr index might assess different nutritious statuses (visceral nutrition versus somatic nutrition), which indicates the implication to assess both indexes independently.

### 4.2. Prognostic Impact of Both Indexes

A group of low GNRI and low modified Cr index had lower survival than the other three combination patterns. Interestingly, if either of them was high, survival was comparable to the group with a high GNRI and high modified Cr index. Although further studies are warranted, the intervention of either of indexes (not necessarily both) might improve survival in patients with hemodialysis.

Both indexes were associated with several causes of death. Tanaka and colleagues similarly demonstrated, using the Q-cohort, that low GNRI was associated with cardiovascular or infection death but not cancer death [14]. Another study found a similar trend in the modified Cr index [11].

A reduced score in GNRI and modified Cr index is associated with incremental mortality [15,16]. Although some small studies observed partial prognostic improvement by the nutritious supports [17,18,19], a methodology of how to improve nutritious status in patients with dialysis has not yet been established. The impact of rehabilitation on somatic nutrition also remains a future concern [20].

### 4.3. Limitations

Some limitations should be noted. First, due to the retrospective nature of our study, all measured or unmeasured confounders may not have been properly controlled. Second, this study consists of a small sample cohort. Third, the optimal cutoff of these indexes remains controversial. A recent study proposed a prognostic cutoff of GNRI as 92, which was approximately comparable to ours.

## 5. Conclusions

A combination of GNRI and modified Cr index may be a promising tool to risk-stratify mortality in patients with hemodialysis.

## Figures and Tables

**Figure 1 nutrients-14-00801-f001:**
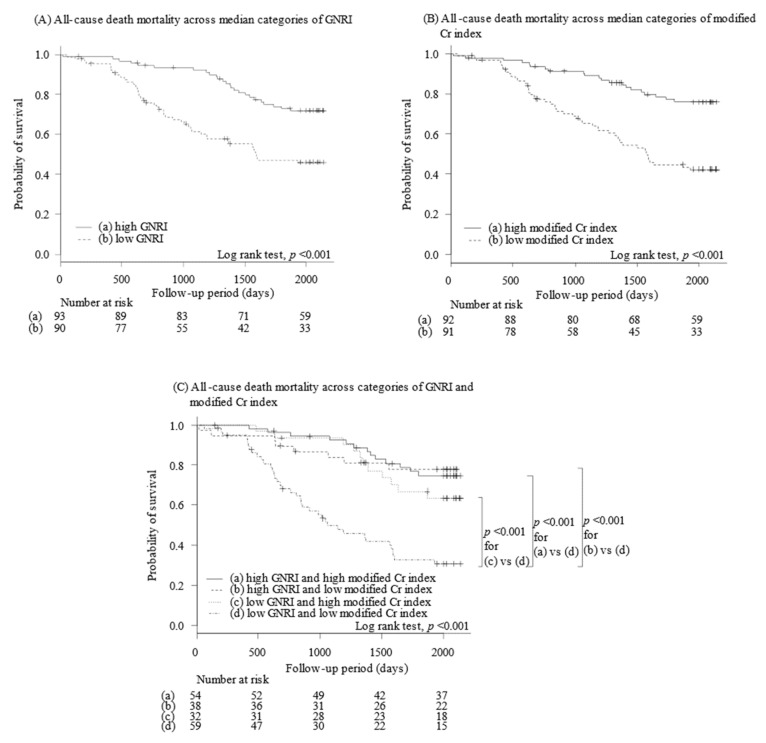
Five-year survival stratified by GNRI (**A**), modified Cr index (**B**), and a combination of both (**C**). Cr, creatinine; GNRI, geriatric nutritional risk index.

**Figure 2 nutrients-14-00801-f002:**
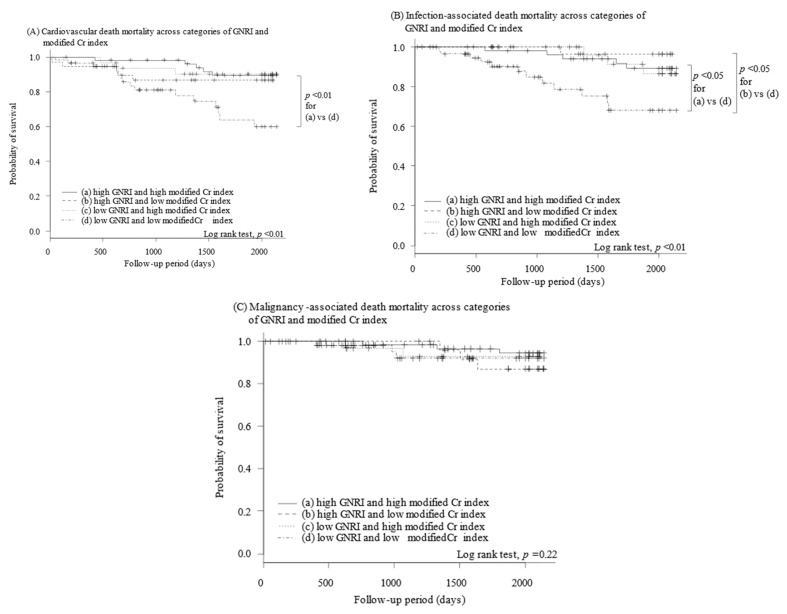
Five-year freedom free from cardiovascular death (**A**), infection death (**B**), and cancer death (**C**) stratified by a combination of GNRI and modified Cr index. Cr, creatinine; GNRI, geriatric nutritional risk index.

**Table 1 nutrients-14-00801-t001:** Baseline characteristics.

	N = 183
Demographics	
Males, *n* (%)	98 (53.6)
Age, years	68.3 ± 12.4
Dialysis vintage, month	97 ± 89
Systolic blood pressure, mmHg	150 ± 25
Comorbidity	
Presence of diabetes, *n* (%)	76 (41)
History of cardiovascular events, *n* (%)	51 (27)
Charlson risk index	3.5 ± 1.3
Cause of end-stage renal disease	
Diabetic nephropathy, *n* (%)	74 (40.4)
Chronic glomerulonephritis, *n* (%)	48 (26.2)
Glomerulosclerosis, *n* (%)	48 (26.2)
Polycystic disease, *n* (%)	16 (8.7)
Others, *n* (%)	16 (8.7)
Laboratory data	
Hemoglobin, g/dL	10.0 ± 1.2
Serum albumin, g/dL	3.4 ± 0.4
Serum urea nitrogen, mg/dL	60.4 ± 14.5
Serum creatinine, mg/dL	10.6 ± 2.6
Serum uric acid, mg/dL	7.6 ± 1.2
Corrected serum calcium, mg/dL	9.3 ± 0.7
Serum phosphate, mg/dL	5.0 ± 1.2
Serum C-reactive protein, mg/dL	0.4 ± 0.7
Serum total cholesterol, mg/dL	148 ± 29
Serum triglycerides, mg/dL	109 ± 53
Intact parathormone, pg/mL	155 ± 167
Kt/V ratio for urea	1.43 ± 0.28
Index	
GNRI	91.2 ± 10.9
Modified Cr index (male), mg/kg/day	22.2 ± 2.5
Modified Cr index (female), mg/kg/day	19.6 ± 2.2
Medication	
Angiotensin receptor blocker, *n* (%)	81 (48)
Calcium channel blocker, *n* (%)	101 (60)
Beta-blocker, *n* (%)	34 (20)
Statin, *n* (%)	25 (15)
Anti-platelets, *n* (%)	82 (49)

Cr, creatinine; GNRI, geriatric nutritional risk index.

**Table 2 nutrients-14-00801-t002:** Prognostic impacts of each index.

	Unadjusted Model		Multivariable Model 1		Multivariable Model 2		Multivariable Model 3	
	Hazard Ratio (95% CI)	*p*-Value	Hazard Ratio (95% CI)	*p*-Value	Hazard Ratio (95% CI)	*p*-Value	Hazard Ratio (95% CI)	*p*-Value
GNRI								
High GNRI	1.00 (reference)	-	1.00 (reference)	-	1.00 (reference)	-	1.00 (reference)	-
Low GNRI	2.59 (1.59–4.23)	<0.001	2.18 (1.31–3.64)	<0.01	2.19 (1.29–3.73)	<0.01	2.05 (1.17–3.60)	<0.05
Modified Cr index								
High modified Cr index	1.00 (reference)	-	1.00 (reference)	-	1.00 (reference)	-	1.00 (reference)	-
Low modified Cr index	3.15 (1.89–5.26)	<0.001	2.02 (1.11–3.68)	<0.05	1.81 (0.97–3.35)	0.06	1.82 (0.94–3.50)	0.07

Unadjusted and multivariable-adjusted HRs were analyzed by the Cox proportional hazards risk model with all-cause death. Multivariable-adjusted model 1 was adjusted for age and sex. Model 2 was adjusted for age, sex, dialysis vintage, the presence of diabetes and history of cardiovascular events. Model 3 was adjusted for age, sex, dialysis vintage, hemoglobin, log_10_ C-reactive protein, phosphate, total cholesterol, intact parathormone, the presence of diabetes, history of cardiovascular events and Charlson risk index. A two-tailed *p*-value < 0.05 was considered statistically significant. High GNRI means GNRI ≥ 91.6. High Cr index means Cr index ≥22.3 for males or ≥19.9 for females. CI, confidence interval; Cr, creatinine; GNRI, geriatric nutritional risk index.

**Table 3 nutrients-14-00801-t003:** Number of patients who were assigned to high/low indexes.

	Men	Women	Total
High GNRI and high modified Cr index, *n* (%)	32 (32.7)	22 (25.9)	54 (29.5)
High GNRI and low modified Cr index, *n* (%)	24 (24.5)	14 (16.5)	38 (20.8)
Low GNRI and high modified Cr index, *n* (%)	16 (16.3)	16 (18.8)	32 (17.5)
Low GNRI and low modified Cr index, *n* (%)	26 (26.5)	33 (38.8)	59 (32.2)

High GNRI means GNRI ≥91.6. High Cr index means Cr index ≥22.3 for males or ≥19.9 for females. Cr, creatinine; GNRI, geriatric nutritional risk index.

**Table 4 nutrients-14-00801-t004:** Prognostic impacts of combination of both indexes.

	Unadjusted Model		Multivariable Model 1		Multivariable Model 2		Multivariable Model 3	
	Hazard Ratio (95% CI)	*p*-Value	Hazard Ratio (95% CI)	*p*-Value	Hazard Ratio (95% CI)	*p*-Value	Hazard Ratio (95% CI)	*p*-Value
All-cause death								
High GNRI and high modified Cr index	1.00 (reference)	-	1.00 (reference)	-	1.00 (reference)	-	1.00 (reference)	-
High GNRI and low modified Cr index	1.48 (0.66–3.29)	0.34	0.92 (0.40–2.11)	0.84	0.73 (0.31–1.78)	0.62	0.69 (0.27–1.71)	0.42
Low GNRI and high modified Cr index	1.04 (0.42–2.55)	0.93	1.11 (0.45–2.72)	0.83	1.06 (0.43–2.61)	0.9	0.75 (0.29–1.93)	0.55
Low GNRI and low modified Cr index	4.60 (2.52–8.39)	<0.001	2.83 (1.41–5.69)	<0.01	2.54 (1.24–5.21)	<0.05	2.33 (1.06–5.13)	<0.05
Cardiovascular death								
High GNRI and high modified Cr index	1.00 (reference)	-	1.00 (reference)	-	1.00 (reference)	-	1.00 (reference)	-
High GNRI and low modified Cr index	1.54 (0.45–5.32)	0.50	1.43 (0.41–4.94)	0.57	1.41 (0.41–4.91)	0.59	1.49 (0.41–5.39)	0.54
Low GNRI and high modified Cr index	1.07 (0.26–4.49)	0.92	0.87 (0.20–3.86)	0.86	0.69 (0.15–3.14)	0.63	0.69 (0.14–3.37)	0.65
Low GNRI and low modified Cr index	4.30 (1.55–11.9)	<0.01	3.46 (1.07–11.1)	<0.05	3.08 (0.91–10.4)	0.07	3.14 (0.89–11.0)	0.07
Infection-associated death								
High GNRI and high modified Cr index	1.00 (reference)	-	1.00 (reference)	-	1.00 (reference)	-	1.00 (reference)	-
High GNRI and low modified Cr index	0.31 (0.04–2.69)	0.29	0.28 (0.03–2.42)	0.25	0.25 (0.03–2.13)	0.20	0.21 (0.02–2.01)	0.17
Low GNRI and high modified Cr index	1.08 (0.26–4.52)	0.92	0.80 (0.18–3.59)	0.76	0.70 (0.15–3.28)	0.65	1.35 (0.24–7.73)	0.74
Low GNRI and low modified Cr index	3.95 (1.38–11.3)	<0.05	3.23 (0.98–10.7)	0.05	2.99 (0.89–9.98)	0.08	8.45 (1.80–39.7)	<0.01

Unadjusted and multivariable-adjusted HRs were analyzed by the Cox proportional hazards risk model with all-cause death. Multivariable-adjusted model 1 was adjusted for age and sex. Model 2 was adjusted for age, sex, dialysis vintage, the presence of diabetes and history of cardiovascular events. Model 3 was adjusted for age, sex, dialysis vintage, hemoglobin, log_10_ C-reactive protein, phosphate, total cholesterol, intact parathormone, the presence of diabetes, history of cardiovascular events and Charlson risk index. A two-tailed *p*-value < 0.05 was considered statistically significant. High GNRI means GNRI ≥ 91.6. High Cr index means Cr index ≥22.3 for males or ≥19.9 for females. Cr, creatinine; GNRI, geriatric nutritional risk index.

## Data Availability

The data used for analysis in the present study are available upon request to the corresponding author. The data are not publicly available because of privacy or ethical restrictions.

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
