# Peer review of "Impact of Geriatric Nutritional Risk Index and Modified Creatinine Index Combination on Mortality in Hemodialysis Patients"

_nutrients, 2022, doi:10.3390/nu14040801_

Round 1
Reviewer 1 Report
Additional discussion is awaited, especially regarding the different aspect of these two measurements. Interestingly, you showed that the two scale are not parallel. However, you didn’t discuss the reason of these differences, and possibly the different pathophysiological aspect of either measurements.
Please use a citation to support the formula, like to Bouillanne O et al., (2005) for GNRI and Arase H et al., (2020) regarding sCr index.
In table 1, please clarify the units. For example, regarding: « Male, % - 98 (53.6) ». The two values are not percentage but a count with the respective percentage. Prefer « Males, n (%) ».
For the shake of clarity, the table 3 deserve percentage
Line 190 and in the conclusion, the term “aggressive” and the meaning of the sentence should be reworded.
An English proofreading is needed.
Reviewer 2 Report
Concerning reviewed paper I have some serious notes.
- In the characteristics of the patients, it is necessary to add the information regarding
the concentration of cholesterol (also the nutritional marker in dialyzed patients) and PTH (secondary hyperthyroidism is associated with a higher risk of cardiovascular complications).
- It is also worth analyzing the relationship between cholesterol and PTH levels and mortality.
- Cholesterol and PTH level should also be used in multivariate analysis models.
- It is worth objectively assessing patients' comorbidity (e.g. Charslon index) and fragility score
- It seems necessary to evaluate the impact of comorbidity, fragility score on mortality, GNRI and the Modified Creatinine Index Combination values.
- Were the patients on a special diet and whether it was assessed at all?
Round 2
Reviewer 2 Report
I recommend to add the conclusion :
The prospective study based on bigger population is necessary to confirm the usage the analyzed parameters.